# Automatic diagnosis of the 12-lead ECG using a deep neural network

Antônio H. Ribeiro [1,2✉], Manoel Horta Ribeiro[1], Gabriela M. M. Paixão[1,3], Derick M. Oliveira[1], Paulo R. Gomes[1,3], Jéssica A. Canazart[1,3], Milton P. S. Ferreira[1], Carl R. Andersson[2], Peter W. Macfarlane[4], Meira Wagner Jr.[1], Thomas B. Schön [2✉] & Antonio Luiz P. Ribeiro [1,3✉]

The role of automatic electrocardiogram (ECG) analysis in clinical practice is limited by the accuracy of existing models. Deep Neural Networks (DNNs) are models composed of stacked transformations that learn tasks by examples. This technology has recently achieved striking success in a variety of task and there are great expectations on how it might improve clinical practice. Here we present a DNN model trained in a dataset with more than 2 million labeled exams analyzed by the Telehealth Network of Minas Gerais and collected under the scope of the CODE (Clinical Outcomes in Digital Electrocardiology) study. The DNN out-perform cardiology resident medical doctors in recognizing 6 types of abnormalities in 12-lead ECG recordings, with F1 scores above 80% and specificity over 99%. These results indicate ECG analysis based on DNNs, previously studied in a single-lead setup, generalizes well to 12-lead exams, taking the technology closer to the standard clinical practice.

[1] Universidade Federal de Minas Gerais, Belo Horizonte, Brazil. [2] Uppsala University, Uppsala, Sweden. [3] Telehealth Center from Hospital das Clínicas da Universidade Federal de Minas Gerais, Belo Horizonte, Brazil. [4] University of Glasgow, Glasgow, Scotland. ✉email: antonio-ribeiro@ufmg.br; thomas. schon@it.uu.se; antonio.ribeiro@ebserh.gov.br

Cardiovascular diseases are the leading cause of death worldwide[1] and the electrocardiogram (ECG) is a major tool in their diagnoses. As ECGs transitioned from analog to digital, automated computer analysis of standard 12-lead electrocardiograms gained importance in the process of medical diagnosis[2,3]. However, limited performance of classical algorithms[4,5] precludes its usage as a standalone diagnostic tool and relegates them to an ancillary role[3,6].

Deep neural networks (DNNs) have recently achieved striking success in tasks such as image classification[7] and speech recognition[8], and there are great expectations when it comes to how this technology may improve health care and clinical practice[9–11]. So far, the most successful applications used a supervised learning setup to automate diagnosis from exams. Supervised learning models, which learn to map an input to an output based on example input−output pairs, have achieved better performance than a human specialist on their routine work-flow in diagnosing breast cancer[12] and detecting retinal diseases from three-dimensional optical coherence tomography scans[13]. While efficient, training DNNs in this setup introduces the need for large quantities of labeled data which, for medical applications, introduce several challenges, including those related to confidentiality and security of personal health information[14].

A convincing preliminary study of the use of DNNs in ECG analysis was recently presented in ref. [15]. For single-lead ECGs, DNNs could match state-of-the-art algorithms when trained in openly available datasets (e.g. 2017 PhysioNet Challenge data[16]) and, for a large enough training dataset, present superior performance when compared to practicing cardiologists. However, as pointed out by the authors, it is still an open question if the application of this technology would be useful in a realistic clinical setting, where 12-lead ECGs are the standard technique[15].

The short-duration, standard, 12-lead ECG (S12L-ECG) is the most commonly used complementary exam for the evaluation of the heart, being employed across all clinical settings, from the primary care centers to the intensive care units. While long-term cardiac monitoring, such as in the Holter exam, provides information mostly about cardiac rhythm and repolarization, the S12L-ECG can provide a full evaluation of the cardiac electrical activity. This includes arrhythmias, conduction disturbances, acute coronary syndromes, cardiac chamber hypertrophy and enlargement and even the effects of drugs and electrolyte disturbances. Thus, a deep learning approach that allows for accurate interpretation of S12L-ECGs would have the greatest impact.

S12L-ECGs are often performed in settings, such as in primary care centers and emergency units, where there are no specialists to analyze and interpret the ECG tracings. Primary care and emergency department health professionals have limited diagnostic abilities in interpreting S12-ECGs[17,18]. The need for an accurate automatic interpretation is most acute in low and middle-income countries, which are responsible for more than 75% of deaths related to cardiovascular disease[19], and where the population, often, do not have access to cardiologists with full expertise in ECG diagnosis.

The use of DNNs for S12L-ECG is still largely unexplored. A contributing factor for this is the shortage of full digital S12L-ECG databases, since most recordings are still registered only on paper, archived as images, or stored in PDF format[20]. Most available databases comprise a few hundreds of tracings and no systematic annotation of the full list of ECG diagnoses[21], limiting their usefulness as training datasets in a supervised learning setting. This lack of systematically annotated data is unfortunate, as training an accurate automatic method of diagnosis from S12L-ECG would be greatly beneficial.

In this paper, we demonstrate the effectiveness of DNNs for automatic S12L-ECG classification. We build a large-scale dataset of labeled S12L-ECG exams for clinical and prognostic studies (the CODE—Clinical Outcomes in Digital Electrocardiology study) and use it to develop a DNN to classify six types of ECG abnormalities considered representative of both rhythmic and morphologic ECG abnormalities.

## Results

**Model specification and training.** We collected a dataset consisting of 2,322,513 ECG records from 1,676,384 different patients of 811 counties in the state of Minas Gerais/Brazil from the Telehealth Network of Minas Gerais (TNMG)[22]. The dataset characteristics are summarized in Table 1. The acquisition and annotation procedures of this dataset are described in Methods. We split this dataset into a training set and a validation set. The training set contains 98% of the data. The validation set consists of the remaining 2% (~50,000 exams) of the dataset and it was used for hyperparameter tuning.

We train a DNN to detect: 1st degree AV block (1dAVb), right bundle branch block (RBBB), left bundle branch block (LBBB), sinus bradycardia (SB), atrial fibrillation (AF) and sinus tachycardia (ST). These six abnormalities are displayed in Fig. 1.

We used a DNN architecture known as the residual network[23], commonly used for images, which we here have adapted to unidimensional signals. A similar architecture has been successfully employed for detecting abnormalities in single-lead ECG signals[15]. Furthermore, in the 2017 Physionet challenge[16], algorithms for detecting AF have been compared in an open dataset of single-lead ECGs and both the architecture described in ref. [15] and other convolutional architectures[24,25] have achieved top scores.

The DNN parameters were learned using the training dataset and our design choices were made in order to maximize the performance on the validation dataset. We should highlight that, despite using a significantly larger training dataset, we got the best validation results with an architecture with, roughly, one quarter the number of layers and parameters of the network employed in ref. [15].

**Testing and performance evaluation.** For testing the model we employed a dataset consisting of 827 tracings from distinct patients annotated by three different cardiologists with experience in electrocardiography (see Methods). The test dataset characteristics are summarized in Table 1. Table 2 shows the performance of the DNN on the test set. High-performance

**Table 1 (Dataset summary) Patient characteristics and abnormalities prevalence, n (%).**

|  | Train + Val (n = 2,322,513) | Test (n = 827) |
|---|---|---|
| Abnormality | | |
| 1dAVb | 35,759 (1.5%) | 28 (3.4%) |
| RBBB | 63,528 (2.7%) | 34 (4.1%) |
| LBBB | 39,842 (1.7%) | 30 (3.6%) |
| SB | 37,949 (1.6%) | 16 (1.9%) |
| AF | 41,862 (1.8%) | 13 (1.6%) |
| ST | 49,872 (2.1%) | 36 (4.4%) |
| Age group | | |
| 16−25 | 155,531 (6.7%) | 43 (5.2%) |
| 26−40 | 406,239 (17.5%) | 122 (14.8%) |
| 41−60 | 901.456 (38.8%) | 340 (41.1%) |
| 61−80 | 729,300 (31.4%) | 278 (33.6%) |
| ≥81 | 129,987 (5.6%) | 44 (5.3%) |
| Sex | | |
| Male | 922,780 (39.7%) | 321 (38.8%) |
| Female | 1,399,733 (60.3%) | 506 (61.2%) |

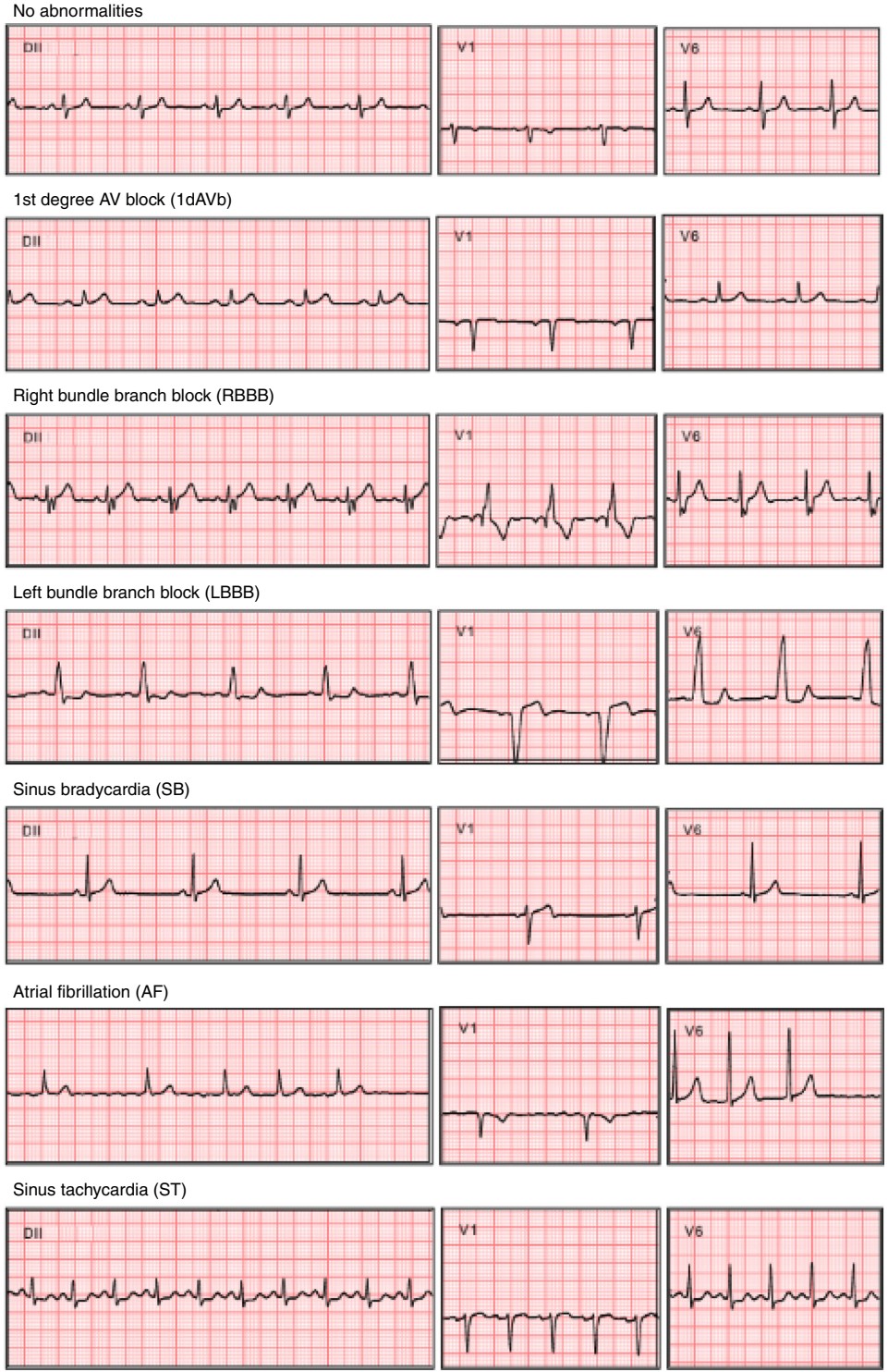

**Fig. 1 Abnormalities examples.** A list of all the abnormalities the model classifies. We show only three representative leads (DII, V1 and V6).

measures were obtained for all ECG abnormalities, with F1 scores above 80% and specificity indexes over 99%. We consider our model to have predicted the abnormality when its output—a number between 0 and 1—is above a threshold. Figure 2 shows the precision-recall curve for our model, for different values of this threshold.

Neural networks are initialized randomly, and different initialization usually yield different results. In order to show the stability of the method, we have trained ten neural networks with the same set of hyperparameters and different initializations. The

range between the maximum and minimum precision among these realizations, for different values of threshold, are the shaded regions displayed in Fig. 2. These realizations have micro average precision (mAP) between 0.946 and 0.961; we choose the one with mAP immediately above the median value of all executions (the one with mAP = 0.951) (We couldn't choose the model with mAP equal to the median value because 10 is an even number; hence, there is no single middle value.). All the analyses from now on will be for this realization of the neural network, which correspond both to the strong line in Fig. 2 and to the scores

**Table 2 (Performance indexes)** Scores of our DNN are compared on the test set with the average performance of: (i) 4th year cardiology resident (cardio.); (ii) 3rd year emergency resident (emerg.); and (iii) 5th year medical students (stud.).

| | Precision (PPV) | | | | Recall (Sensitivity) | | | | Specificity | | | | F1 score | | | |
|---|---|---|---|---|---|---|---|---|---|---|---|---|---|---|---|---|
| | DNN | cardio. | emerg. | stud. | DNN | cardio. | emerg. | stud. | DNN | cardio. | emerg. | stud. | DNN | cardio. | emerg. | stud. |
| 1dAVb | 0.867 | 0.905 | 0.639 | 0.605 | 0.929 | 0.679 | 0.821 | 0.929 | 0.995 | 0.997 | 0.984 | 0.979 | **0.897** | 0.776 | 0.719 | 0.732 |
| RBBB | 0.895 | 0.868 | 0.963 | 0.914 | 1.000 | 0.971 | 0.765 | 0.941 | 0.995 | 0.994 | 0.999 | 0.996 | **0.944** | 0.917 | 0.852 | 0.928 |
| LBBB | 1.000 | 1.000 | 0.963 | 0.931 | 1.000 | 0.900 | 0.867 | 0.900 | 1.000 | 1.000 | 0.999 | 0.997 | **1.000** | 0.947 | 0.912 | 0.915 |
| SB | 0.833 | 0.833 | 0.824 | 0.750 | 0.938 | 0.938 | 0.875 | 0.750 | 0.996 | 0.996 | 0.996 | 0.995 | **0.882** | **0.882** | 0.848 | 0.750 |
| AF | 1.000 | 0.769 | 0.800 | 0.571 | 0.769 | 0.769 | 0.615 | 0.923 | 1.000 | 0.996 | 0.998 | 0.989 | **0.870** | 0.769 | 0.696 | 0.706 |
| ST | 0.947 | 0.968 | 0.946 | 0.912 | 0.973 | 0.811 | 0.946 | 0.838 | 0.997 | 0.999 | 0.997 | 0.996 | **0.960** | 0.882 | 0.946 | 0.873 |

*PPV* positive predictive value. The bold values represent the best scores.

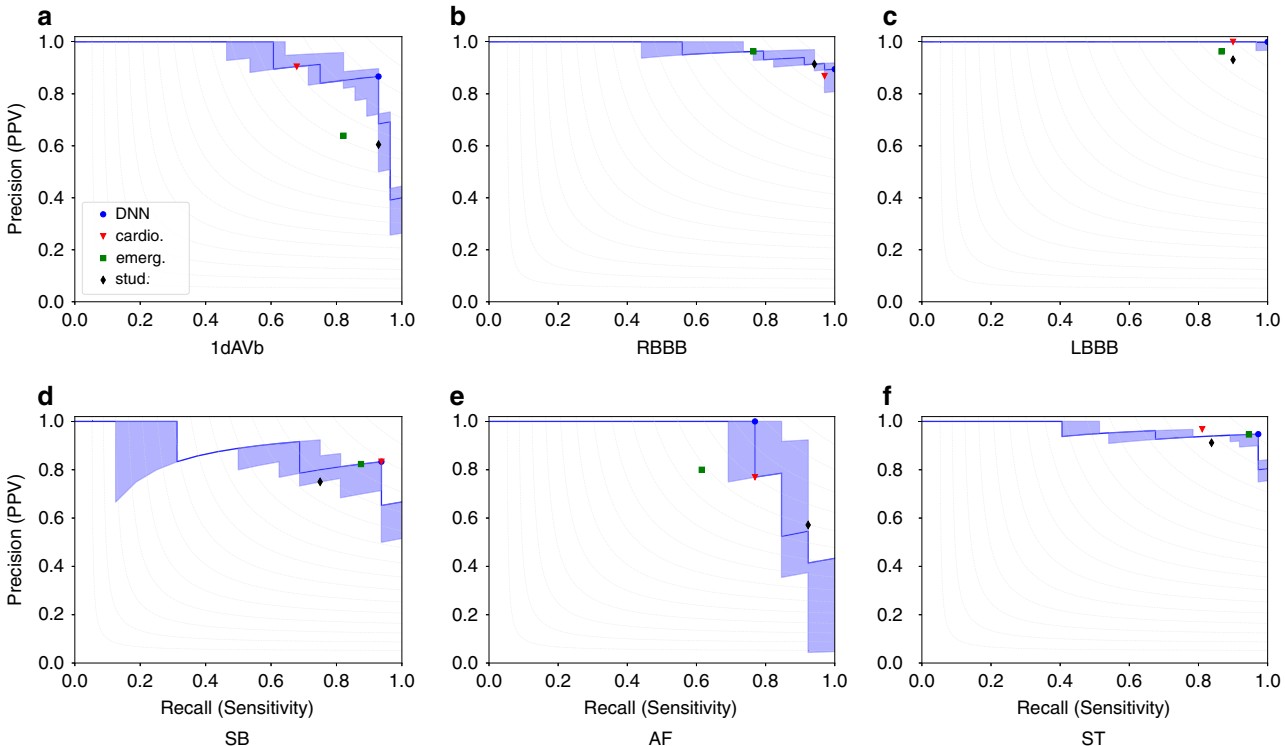

**Fig. 2 Precision-recall curve.** Show precision-recall curve for our nominal prediction model on the test set (strong line) with regard to each ECG abnormalities. The shaded region shows the range between maximum and minimum precision for neural networks trained with the same configuration and different initialization. Points corresponding to the performance of resident medical doctors and students are also displayed, together with the point corresponding to the DNN performance for the same threshold used for generating Table 2. Gray dashed curves in the background correspond to iso-$F_1$ curves (i.e. curves in the precision-recall plane with constant F1 score).

presented in Table 2. For this model, Fig. 2 shows the point corresponding to the maximum F1 score for each abnormality. The threshold corresponding to this point is used for producing the DNN scores displayed in Table 2.

The same dataset was evaluated by: (i) two 4th year cardiology residents; (ii) two 3rd year emergency residents; and (iii) two 5th year medical students. Each one annotated half of the exams in the test set. Their average performances are given, together with the DNN results, in Table 2 and their precision-recall scores are plotted in Fig. 2. Considering the F1 score, the DNN matches or outperforms the medical residents and students for all abnormalities. The confusion matrices and the inter-rater agreement (kappa coefficients) for the DNN, the resident medical doctors and students are provided, respectively, in Supplementary Tables 1 and 2(a). Additionally, in Supplementary Table 2(b), we compare the inter-rater agreement between the neural network and the certified cardiologists that annotated the test set.

A trained cardiologist reviewed all the mistakes made by the DNN, the medical residents and the students, trying to explain the source of the error. The cardiologist had meetings with the residents and students where they together agreed on which was the source of the error. The results of this analysis are given in Table 3.

In order to compare the performance difference between the DNN and resident medical doctors and students, we compute empirical distributions for the precision (PPV), recall (sensitivity), specificity and F1 score using bootstrapping[26]. The boxplots corresponding to these bootstrapped distributions are presented in Supplementary Fig. 1. We have also applied the McNemar test[27] to compare the misclassification distribution of the DNN, the medical residents and the students. Supplementary Table 3 shows the $p$ values of the statistical test. Both analyses do not indicate a statistically significant difference in performance among the DNN and the medical residents and students for most of the classes.

**Table 3 (Error analysis) Present the analysis of misclassified exams.**

| | DNN | | | cardio. | | | | emerg. | | | | stud. | | | |
|---|---|---|---|---|---|---|---|---|---|---|---|---|---|---|---|
| | meas. | noise | unexplain. | meas. | noise | concep. | atte. | meas. | noise | concep. | atte. | meas. | noise | concep. | atte. |
| 1dAVb | 3 | 2 | 1 | 8 | 3 | | | 15 | 3 | | | 13 | 3 | 3 | |
| RBBB | 3 | | 1 | 4 | | 2 | | 1 | | 8 | | 3 | | 2 | |
| LBBB | | | | 1 | 1 | 1 | | | 1 | 4 | | | 2 | 3 | |
| SB | 4 | | | 4 | | | | 4 | | | 1 | 5 | | 2 | 1 |
| AF | | 2 | 1 | | 4 | 2 | | | 2 | 5 | | | 3 | 7 | |
| ST | 2 | | 1 | 2 | 1 | | 5 | 1 | 1 | 1 | 1 | 1 | 2 | 1 | 5 |

The errors were classified into the following categories: (i) measurements errors (meas.) were ECG interval measurements preclude the given diagnosis by its textbook definition; (ii) errors due to noise, where we believe that the analyst or the DNN failed due to a lower than usual signal quality; and (iii) other type of errors (unexplain.). Those were further divided, for the medical residents and students, into two categories: conceptual errors (concep.), where our reviewer suggested that the doctor failed to understand the definitions of each abnormality, and attention errors (atte.), where we believe the error could be avoided if the reviewer had been more careful.

Finally, to assess the effect of how we structure our problem, we have considered alternative scenarios where we use the 2,322,513 ECG records in 90%5%-5% splits, stratified randomly, by patient or in chronological order. Being the splits used, respectively, for training, validation and as a second larger *test* set. The results indicate no statistically significant difference between the original DNN used in our analysis and the alternative models developed in the 90%-5%-5% splits. The exception is the model developed using the chronologically order split, for which the changes along time in the telehealth center operation have affected the splits (cf. Supplementary Fig. 2).

## Discussion

This paper demonstrates the effectiveness of "end-to-end" automatic S12L-ECG classification. This presents a paradigm shift from the classical ECG automatic analysis methods[28]. These classical methods, such as the University of Glasgow ECG analysis program[29], first extract the main features of the ECG signal using traditional signal processing techniques and then use these features as inputs to a classifier. End-to-end learning presents an alternative to these two-step approaches, where the raw signal itself is used as an input to the classifier which learns, by itself, to extract the features. This approach has presented, in a emergency room setting, performance superior to commercial ECG software based on traditional signal processing techniques[30].

Neural networks have previously been used for classification of ECGs both in a classical—feature-based—setup[31,32] and in an end-to-end learn setup[15,33,34]. Hybrid methods combining the two paradigms are also available: the classification may be done using a combination of handcrafted and learned features[35] or by using a two-stage training, obtaining one neural network to learn the features and another to classify the exam according to these learned features[36].

The paradigm shift towards end-to-end learning had a significant impact on the size of the datasets used for training the models. Many results using classical methods[28,34,36] train their models on datasets with few examples, such as the MIT-BIH arrhythmia database[37], with only 47 unique patients. The most convincing papers using end-to-end deep learning or mixed approaches, on the other hand, have constructed large datasets, ranging from 3000 to 100,000 unique patients, for training their models[15,16,30,35].

Large datasets from previous work[15,16,35], however, either were obtained from cardiac monitors and Holter exams, where patients are usually monitored for several hours and the recordings are restricted to one or two leads or consist of 12-lead ECGs obtained in an emergency room setting[30,38]. Our dataset with well over 2 million entries, on the other hand, consists of short duration (7−10 s) S12L-ECG tracings obtained from in-clinic exams and is

orders of magnitude larger than those used in previous studies. It encompasses not only rhythm disorders, like AF, SB and ST, as in previous studies[15], but also conduction disturbances, such as 1dAVb, RBBB and LBBB. Instead of beat-to-beat classification, as in the MIT-BIH arrhythmia database, our dataset provides annotation for S12L-ECG exams, which are the most common in clinical practice.

The availability of such a large database of S12L-ECG tracings, with annotation for the whole spectrum of ECG abnormalities, opens up the possibility of extending initial results of end-to-end DNN in ECG automatic analysis[15] to a system with applicability in a wide range of clinical settings. The development of such technologies may yield high-accuracy automatic ECG classification systems that could save clinicians considerable time and prevent wrong diagnoses. Millions of S12L-ECGs are performed every year, many times in places where there is a shortage of qualified medical doctors to interpret them. An accurate classification system could help to detect wrong diagnoses and improve the access of patients from deprived and remote locations to this essential diagnostic tool of cardiovascular diseases.

The error analysis shows that most of the DNN mistakes were related to measurements of ECG intervals. Most of those were borderline cases, where the diagnosis relies on a consensus definitions[39] that can only be ascertained when a measurement is above a sharp cutoff point. The mistakes can be explained by the DNN failing to encode these very sharp thresholds. For example, the DNN wrongly detecting an SB with a heart rate slightly above 50 bpm or an ST with a heart rate slightly below 100 bpm. Supplementary Fig. 3 illustrates this effect. Noise and interference in the baseline are established causes of error[40] and affected both automatic and manual diagnosis of ECG abnormalities. Nevertheless, the DNN seems to be more robust to noise and it made fewer mistakes of this type compared to the medical residents and students. Conceptual errors (where our reviewer suggested that the doctor failed to understand the definitions of each abnormality) were more frequent for emergency residents and medical students than for cardiology residents. Attention errors (where we believe that the error could have been avoided if the manual reviewer were more careful) were present at a similar ratio for cardiology residents, emergency residents and medical students.

Interestingly, the performance of the emergency residents is worse than the medical students for many abnormalities. This might seem counter-intuitive because they have less years of medical training. It might, however, be justified by the fact that emergency residents, unlike cardiology residents, do not have to interpret these exams on a daily basis, while medical students still have these concepts fresh from their studies.

Our work is perhaps best understood in the context of its limitations. While we obtained the highest F1 scores for the DNN, the McNemar statistical test and bootstrapping suggest that we do

not have confidence enough to assert that the DNN is actually better than the medical residents and students with statistical significance. We attribute this lack of confidence in the comparison to the presence of relatively infrequent classes, where a few erroneous classifications may significantly affect the scores. Furthermore, we did not test the accuracy of the DNN in the diagnosis of other classes of abnormalities, like those related to acute coronary syndromes or cardiac chamber enlargements and we cannot extend our results to these untested clinical situations. Indeed, the real clinical setting is more complex than the experimental situation tested in this study and, in complex and borderline situations, ECG interpretation can be extremely difficult and may demand the input of highly specialized personnel. Thus, even if a DNN is able to recognize typical ECG abnormalities, further analysis by an experienced specialist will continue to be necessary to these complex exams.

This proof-of-concept study, showing that a DNN can accurately recognize ECG rhythm and morphological abnormalities in clinical S12L-ECG exams, opens a series of perspectives for future research and clinical applications. A next step would be to prove that a DNN can effectively diagnose multiple and complex ECG abnormalities, including myocardial infarction, cardiac chamber enlargement and hypertrophy and less common forms of arrhythmia, and to recognize a normal ECG. Subsequently, the algorithm should be tested in a controlled real-life situation, showing that accurate diagnosis could be achieved in real time, to be reviewed by clinical specialists with solid experience in ECG diagnosis. This real-time, continuous evaluation of the algorithm would provide rapid feedback that could be incorporated as further improvements of the DNN, making it even more reliable.

The TNMG, the large telehealth service from which the dataset used was obtained[22], is a natural laboratory for these next steps, since it performs more than 2000 ECGs a day and it is currently expanding its geographical coverage over a large part of a continental country (Brazil). An optimized system for ECG interpretation, where most of the classification decisions are made automatically, would imply that the cardiologists would only be needed for the more complex cases. If such a system is made widely available, it could be of striking utility to improve access to health care in low- and middle-income countries, where cardiovascular diseases are the leading cause of death and systems of care for cardiac diseases are lacking or not working well[41].

In conclusion, we developed an end-to-end DNN capable of accurately recognizing six ECG abnormalities in S12L-ECG exams, with a diagnostic performance at least as good as medical residents and students. This study shows the potential of this technology, which, when fully developed, might lead to more reliable automatic diagnosis and improved clinical practice. Although expert review of complex and borderline cases seems to be necessary even in this future scenario, the development of such automatic interpretation by a DNN algorithm may expand the access of the population to this basic and useful diagnostic exam.

## Methods

**Dataset acquisition**. All S12L-ECGs analyzed in this study were obtained by the Telehealth Network of Minas Gerais (TNMG), a public telehealth system assisting 811 out of the 853 municipalities in the state of Minas Gerais, Brazil[22]. Since September 2017, the TNMG has also provided telediagnostic services to other Brazilian states in the Amazonian and Northeast regions. The S12L-ECG exam was performed mostly in primary care facilities using a tele-electrocardiograph manufactured by Tecnologia Eletrônica Brasileira (São Paulo, Brazil)—model TEB ECGPC—or Micromed Biotecnologia (Brasilia, Brazil)—model ErgoPC 13. The duration of the ECG recordings is between 7 and 10 s sampled at frequencies ranging from 300 to 600 Hz. A specific software developed in-house was used to capture the ECG tracings, to upload the exam together with the patient's clinical history and to send it electronically to the TNMG analysis center. Once there, one cardiologist from the TNMG experienced team analyzes the exam and a report is

made available to the health service that requested the exam through an online platform.

We have incorporated the University of Glasgow (Uni-G) ECG analysis program (release 28.5, issued in January 2014) in the in-house software since December 2017. The analysis program was used to automatically identify waves and to calculate axes, durations, amplitudes and intervals, to perform rhythm analysis and to give diagnostic interpretation[29,42]. The Uni-G analysis program also provides Minnesota codes[43], a standard ECG classification used in epidemiological studies[44]. Since April 2018 the automatic measurements are being shown to the cardiologists that give the medical report. All clinical information, digital ECGs tracings and the cardiologist report were stored in a database. All previously stored data were also analyzed by Uni-G software in order to have measurements and automatic diagnosis for all exams available in the database, since the first recordings. The CODE study was established to standardize and consolidate this database for clinical and epidemiological studies. In the present study, the data (for patients above 16 years old) obtained between 2010 and 2016 were used in the training and validation set and, from April to September 2018, in the test set.

**Labeling training data from text report**. For the training and validation sets, the cardiologist report is available only as a textual description of the abnormalities in the exam. We extract the label from this textual report using a three-step procedure. First, the text is preprocessed by removing stop-words and generating n-grams from the medical report. Then, the Lazy Associative Classifier (LAC)[45], trained on a 2800-sample dictionary created from real diagnoses text reports, is applied to the n-grams. Finally, the text label is obtained using the LAC result in a rule-based classifier for class disambiguation. The classification model reported above was tested on 4557 medical reports manually labeled by a certified cardiologist who was presented with the free-text and was required to choose among the prespecified classes. The classification step recovered the true medical label with good results; the macro F1 score achieved were: 0.729 for 1dAVb; 0.849 for RBBB; 0.838 for LBBB; 0.991 for SB; 0.993 for AF; 0.974 for ST.

**Training and validation set annotation**. To annotate the training and validation datasets, we used: (i) the Uni-G statements and Minnesota codes obtained by the Uni-G automatic analysis (automatic diagnosis); (ii) automatic measurements provided by the Uni-G software; and (iii) the text labels extracted from the expert text reports using the semi-supervised methodology (medical diagnosis). Both the automatic and medical diagnosis are subject to errors: automatic classification has limited accuracy[3–6] and text labels are subject to errors of both the practicing expert cardiologists and the labeling methodology. Hence, we combine the expert annotation with the automatic analysis to improve the quality of the dataset. The following procedure is used for obtaining the ground truth annotation:

1. We:
   (a) Accept a diagnosis (consider an abnormality to be present) if both the expert and either the Uni-G statement or the Minnesota code provided by the automatic analysis indicated the same abnormality.
   (b) Reject a diagnosis (consider an abnormality to be absent) if only one automatic classifier indicates the abnormality in disagreement with both the doctor and the other automatic classifier.

   After this initial step, there are two scenarios where we still need to accept or reject diagnoses. They are: (i) both classifiers indicate the abnormality, but the expert does not; or (ii) only the expert indicates the abnormality, whereas none of the classifiers indicates anything.

2. We used the following rules to reject some of the remaining diagnoses:
   (a) Diagnoses of ST where the heart rate was below 100 (8376 medical diagnoses and 2 automatic diagnoses) were rejected.
   (b) Diagnoses of SB where the heart rate was above 50 (7361 medical diagnoses and 16,427 automatic diagnosis) were rejected.
   (c) Diagnoses of LBBB or RBBB where the duration of the QRS interval was below 115 ms (9313 medical diagnoses for RBBB and 8260 for LBBB) were rejected.
   (d) Diagnoses of 1dAVb where the duration of the PR interval was below 190 ms (3987 automatic diagnoses) were rejected.

3. Then, using the sensitivity analysis of 100 manually reviewed exams per abnormality, we came up with the following rules to accept some of the remaining diagnoses:
   (a) For RBBB, d1AVb, SB and ST, we accepted all medical diagnoses. 26,033, 13,645, 12,200 and 14,604 diagnoses were accepted in this fashion, respectively.
   (b) For AF, we required not only that the exam was classified by the doctors as true, but also that the standard deviation of the NN intervals was higher than 646. 14,604 diagnoses were accepted using this rule.

   According to the sensitivity analysis, the number of false positives that would be introduced by this procedure was smaller than 3% of the total number of exams.

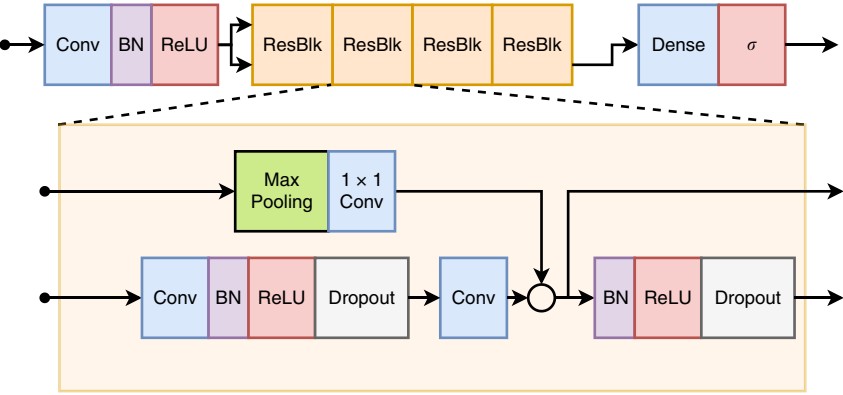

**Fig. 3 (DNN architecture).** The unidimensional residual neural network architecture used for ECG classification.

4. After this process, we were still left with 34,512 exams where the corresponding diagnoses could neither be accepted nor rejected. These were manually reviewed by medical students using the Telehealth ECG diagnostic system, under the supervision of a certified cardiologist with experience in ECG interpretation. The process of manually reviewing these ECGs took several months.

It should be stressed that information from previous medical reports and automatic measurements were used only for obtaining the ground truth for training and validation sets and not on later stages of the DNN training.

**Test set annotation**. The dataset used for testing the DNN was also obtained from TNMG's ECG system. It was independently annotated by two certified cardiologists with experience in electrocardiography. The kappa coefficients[46] indicate the inter-rater agreement for the two cardiologists and are: 0.741 for 1dAVb; 0.955 for RBBB; 0.964 for LBBB; 0.844 for SB; 0.831 for AF; 0.902 for ST. When they agreed, the common diagnosis was considered as ground truth. In cases where there was any disagreement, a third senior specialist, aware of the annotations from the other two, decided the diagnosis. The American Heart Association standardization[47] was used as the guideline for the classification.

It should be highlighted that the annotation was performed in an upgraded version of the TNMG software, in which the automatic measurements obtained by the Uni-G program are presented to the specialist, that has to choose the ECG diagnosis among a number of prespecified classes of abnormalities. Thus, the diagnosis was codified directly into our classes and there was no need to extract the label from a textual report, as it was done for the training and validation sets.

**Neural network architecture and training**. We used a convolutional network similar to the residual network[23], but adapted to unidimensional signals. This architecture allows DNNs to be efficiently trained by including skip connections. We have adopted the modification in the residual block proposed in ref. [48], which place the skip connection in the position displayed in Fig. 3.

All ECG recordings are resampled to a 400 Hz sampling rate. The ECG recordings, which have between 7 and 10 s, are zero-padded resulting in a signal with 4096 samples for each lead. This signal is the input for the neural network.

The network consists of a convolutional layer (Conv) followed by four residual blocks with two convolutional layers per block. The output of the last block is fed into a fully connected layer (Dense) with a sigmoid activation function, $\sigma$, which was used because the classes are not mutually exclusive (i.e. two or more classes may occur in the same exam). The output of each convolutional layer is rescaled using batch normalization, (BN)[49], and fed into a rectified linear activation unit (ReLU). Dropout[50] is applied after the nonlinearity.

The convolutional layers have filter length 16, starting with 4096 samples and 64 filters for the first layer and residual block and increasing the number of filters by 64 every second residual block and subsampling by a factor of 4 every residual block. Max Pooling[51] and convolutional layers with filter length 1 (1x1 Conv) are included in the skip connections to make the dimensions match those from the signals in the main branch.

The average cross-entropy is minimized using the Adam optimizer[52] with default parameters and learning rate lr = 0.001. The learning rate is reduced by a factor of 10 whenever the validation loss does not present any improvement for seven consecutive epochs. The neural network weights was initialized as in ref. [53] and the bias was initialized with zeros. The training runs for 50 epochs with the final model being the one with the best validation results during the optimization process.

**Hyperparameter tuning**. This final architecture and configuration of hyperparameters was obtained after approximately 30 iterations of the procedure: (i) find the neural network weights in the training set; (ii) check the performance in the validation set; and (iii) manually choose new hyperparameters and architecture

using insight from previous iterations. We started this procedure from the set of hyperparameters and architecture used in ref. [15]. It is also important to highlight that the choice of architecture and hyperparameters was done together with improvements in the dataset. Expert knowledge was used to take decision about how to incorporate, on the manual tuning procedure, information about previous iteration that were evaluated on slightly different versions of the dataset.

The hyperparameters were chosen among the following options: residual neural networks with {2, 4, 8, 16} residual blocks, kernel size {8, 16, 32}, batch size {16, 32, 64}, initial learning rate {0.01, 0.001, 0.0001}, optimization algorithms {SGD, ADAM}, activation functions {ReLU, ELU}, dropout rate {0, 0.5, 0.8}, number of epochs without improvement in plateaus between 5 and 10, that would result in a reduction in the learning rate between 0.1 and 0.5. Besides that, we also tried to: (i) use vectorcardiogram linear transformation to reduce the dimensionality of the input; (ii) include LSTM layer before convolutional layers; (iii) use residual network without the preactivation architecture proposed in ref. [48]; (iv) use the convolutional architecture known as VGG; (v) switch the order of activation and batch normalization layer.

**Statistical and empirical analysis of test results**. We computed the precision-recall curve to assess the model discrimination of each rhythm class. This curve shows the relationship between precision (PPV) and recall (sensitivity), calculated using binary decision thresholds for each rhythm class. For imbalanced classes, such as our test set, this plot is more informative than the ROC plot[54]. For the remaining analyses we fixed the DNN threshold to the value that maximized the F1 score, which is the harmonic mean between precision and recall. The F1 score was chosen here due to its robustness to class imbalance[54].

For the DNN with a fixed threshold, and for the medical residents and students, we computed the precision, the recall, the specificity, the F1 score and, also, the confusion matrix. This was done for each class. Bootstrapping[26] was used to analyze the empirical distribution of each of the scores: we generated 1000 different sets by sampling with replacement from the test set, each set with the same number samples as in the test set, and computed the precision, the recall, the specificity and the F1 score for each. The resulting distributions are presented as a boxplot. We used the McNemar test[27] to compare the misclassification distribution of the DNN and the medical residents and students on the test set and the kappa coefficient[46] to compare the inter-rater agreement.

All the misclassified exams were reviewed by an experienced cardiologist and, after an interview with the ECG reviewers, the errors were classified into: measurement errors, noise errors and unexplained errors (for the DNN only) and conceptual and attention errors (for medical residents and students only).

We evaluate the F1 score for alternative scenarios where we use 90%-5%-5% splits of the 2,322,513 records, with the splits ordered randomly, by date, and stratified by patients. The neural networks developed in these alternative scenarios are evaluated on both the original test set ($n = 827$) and the additional test splits (last 5% split). The distribution of the performance in each scenario is computed by a bootstrap analysis (with 1000 and 200 samples, respectively) and the resulting boxplots are displayed in the Supplementary Material.

**Reporting summary**. Further information on research design is available in the Nature Research Reporting Summary linked to this article.

## Data availability

The test dataset used in this study is openly available, and can be downloaded at (https://doi.org/10.5281/zenodo.3625006). The weights of all deep neural network models we developed for this paper are available at (https://doi.org/10.5281/zenodo.3625017). Restrictions apply to the availability of the training set. Requests to access the training data will be considered on an individual basis by the Telehealth Network of Minas Gerais. Any data use will be restricted to noncommercial research purposes, and the data

will only be made available on execution of appropriate data use agreements. The source data underlying Supplementary Figs. 1 and 2 are provided as a Source Data file.

## Code availability

The code for training and evaluating the DNN model, and, also, for generating figures and tables in this paper is available at: https://github.com/antonior92/automatic-ecg-diagnosis.

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

## Acknowledgements

This research was partly supported by the Brazilian Agencies CNPq, CAPES, and FAPEMIG, by projects IATS, MASWeb, INCT-Cyber, Rede de Teleassistência de Minas Gerais and Atmosphere, and by the Wallenberg AI, Autonomous Systems and Software Program (WASP) funded by Knut and Alice Wallenberg Foundation. We also thank NVIDIA for awarding our project with a Titan V GPU. A.L.P.R. receives unrestricted research scholarships from CNPq and FAPEMIG (PPM); W.M.Jr. receives an unrestricted research scholarship from CNPq; A.H.R. receives scholarships from CAPES and CNPq; and, M.H.R. and D.M.O. receive Google Latin America Research Award scholarships. None of the funding agencies had any role in the design, analysis or interpretation of the study. Open access funding provided by Uppsala University.

## Author contributions

A.H.R., M.H.R., G.M.M.P., D.M.O., P.R.G., J.A.C, M.P.S.F and A.L.P.R were responsible for the study design. A.L.P.R conceived the project and acted as project leader. A.H.R., M.H.R and C.R.A. chose the architecture, implemented and tuned the deep neural network. A.H.R did the statistical analysis of the test data and generated the figures and tables. M.H.R., G.M.M.P, J.A.C. were responsible for the preprocessing and annotating the datasets. G.M.M.P was responsible for the error analysis. D.M.O. implemented the semi-supervised methodology to extract the text label. P.R.G. implemented the user interface used to generate the dataset. P.R.G. and M.P.S.F were responsible for maintenance and extraction of the database. P.W.M., W.M.Jr., and T.B.S. helped in the interpretation of the data. A.H.R., M.H.R, P.W.M., T.B.S. and A.L.P.R. contributed to the writing and all authors revised it critically for important intellectual content. All authors read and approved the submitted manuscript.

## Competing interests

The authors declare no competing interests.

## Ethics statement

This study complies with all relevant ethical regulations and was approved by the Research Ethics Committee of the Universidade Federal de Minas Gerais, protocol 49368496317.7.0000.5149. Since this is a secondary analysis of anonymized data stored in the TNMG, informed consent was not required by the Research Ethics Committee for the present study. Researchers signed terms of confidentiality and data utilization.
