## [Peer Review File · Nature Communications]

Reviewers' comments:

Reviewer #1 (Remarks to the Author):

This is a very interesting paper documenting ability of a deep neural network (DNN) to identify abnormalities in a standard 12-lead ECG recordings. The authors have used the largest to date dataset with over 2 million ECGs to train the network, subsequently validating it on about 50,000 ECGs and next testing it on a set of 827 ECGs read by health professionals with different level of expertise. The performance of the DNN was remarkable with the F1 scores (reflecting precision and recall) of 0.85 and specificity index of 0.99.

The methodological approach with so called "end-to-end" automatic classification is what mimics human approach to an ECG image, unlike few prior studies employing ECG signal processing to identify parameters of interest which subsequently were used for automatic classification.

The study employed the state-of-the-art the University fo Glasgow ECG analysis program in combination with Minnesota codes to annotate training and validation sets.

The study focuses on the key clinical ECG abnormalities relevant for daily practice with practical applicability of the system for widespread use in clinics without access to ECG expertise. Although overreading of complex cases is needed, the diagnostic yield of the DNN system is very impressive and deserves implementation on daily practice.

Reviewer #2 (Remarks to the Author):

General comments:

The authors introduce a novel dataset consisting of over 2 million EKG traces, using which they develop a 9-layer ResNet architecture to classify cardiac abnormalities. The performance of that model is presented and claimed to outperform medical students.

Major comments:

Results:

Ref 15 is a poor comparison for this work, apart from the fact that there are a number of questionable and arbitrary design decisions incorporated in that model, it was, similar to this paper designed and trained using a closed source dataset. The authors highlight the 2017 physionet challenge— a reputable venue for such a class of problems, however none of the work are referenced. Notably some of the top ranked work out of that competition use Deep CNN for classifying arrhythmia, similar to this work. I urge the authors to consider the work resulting from that rigorous validation over ref 15 for the motivation for the proposed architecture.

<https://physionet.org/challenge/2017/sources/>

Hong, S., Wu, M., Zhou, Y., Wang, Q., Shang, J., Li, H. and Xie, J., 2017, September. ENCASE: An ENsemble CIASsifiEr for ECG classification using expert features and deep neural networks. In 2017 Computing in Cardiology (CinC) (pp. 1-4). IEEE.

Zabihi, M., Rad, A.B., Katsaggelos, A.K., Kiranyaz, S., Narkilahti, S. and Gabbouj, M., 2017, September. Detection of atrial fibrillation in ECG hand-held devices using a random forest classifier. In 2017 Computing in Cardiology (CinC) (pp. 1-4). IEEE.

Kamaleswaran, R., Mahajan, R. and Akbilgic, O., 2018. A robust deep convolutional neural network for the classification of abnormal cardiac rhythm using single lead electrocardiograms of variable length. *Physiological measurement*, 39(3), p.035006.

Methods:

- An extensive methodological approach is provided for both the training and testing datasets. The training dataset uses a method for automatic classification developed by one of the co-authors (PWM), i.e. the Uni-G. Why was an external EKG classification tool not compared? Can you provide some justification for the significant reliance on this software for label generation?
- There should be better empirical reasoning for why 98% of the dataset was used for training, furthermore why 827 samples were selected for the test.
- There needs to be more details highlighted about the distribution of these classes in the training, validation and test sets.
- While lines 271 through 282 reflect highly optimistic views of the work, there should be some caution when issuing states such as 'state of the art, leading to widespread use etc.' The authors are certainly commended for the work in developing the dataset, and for demonstrating this 'proof of concept' study, however due to a lack of external validation with other arrhythmia detection models these broad statements are premature.

Minor comments:

- Line 300, does a 'team of cardiologist' actually interpret the EKG or is it a single cardiologist? How many inputs are there in the final assessment? This is unclear from the current text.
- Line 334, additional clarity is required, how was the test set labeled? Were there any further validation performed on the classification model? Inclusion of this detail would be great for supplemental material.
- Line 436 Why was zero padding used, were other padding methods experimented?
- Line 447, again supplemental materials on the selection of these hyperparameters would be expected.

Reviewer #3 (Remarks to the Author):

Summary:

The authors present a neural network that can classify a range of ECG abnormalities from short 12-lead ECG recordings and they compare their results with trained individuals.

Major comments:

1) The authors claim in the abstract that "The analysis of the digital ECG [...] have not been studied in an end-to-end machine learning scenario", This is a vast overstatement, as a lot of prior literature exists on this subject. The authors do soften their claims in the introductory section, where they merely claim that "it is still an open question if the application of this technology would be useful in a realistic clinical setting, where 12-lead ECGs are the standard technique". It is worth pointing out that

a number of other published work has applied Deep Learning to 12-lead ECG (albeit some of this might be considered concurrent work) [1-3]. However, the dataset proposed in this study is of larger size than in [1-3], which is why I think the current work is of interest once the other comments are addressed.

2) The authors raise the biggest concern of the underlying work themselves:

"While we obtained the highest F1 scores for the DNN, the McNemar statistical test and bootstrapping suggest that we do not have confidence enough to assert that the DNN is actually better than the medical residents and students with statistical significance." (line 224)

3) There is a very high discrepancy between number of training and test samples (0.03%).

A typical split would be 8:1:1 for training, validation and test.

Having an additional test set to compare to experts is desirable but currently this set is tiny and the values in the results tables only stem from one cardiology resident, one emergency resident and one medical student. Also note that the F1 scores in Table 1 are on average higher than these of the emergency resident.

The authors should provide more details on that.

4) There is an automatic system in place which provides Minnesota codes.

The result of this system should be used as a baseline comparison method.

5) Insufficient description of the model selection process.

The authors used a very similar network architecture as in [4] but they do not describe how they selected their final network. In particular,

it is unclear which hyperparameters were considered, e.g. which learning rates, optimizers, activation functions (Relu, leaky [5], SELU [6]), number of neurons, number of layers, kernel sizes, etc were considered at all. Furthermore, the authors do not describe how the hyperparameters were selected. Has this been done by manual hyperparameter selection, grid-search or by model-based approach (see hyperparameter selection strategies given in ref [7])?

The authors should provide details, which hyperparameters were considered and how they were selected together with performance comparisons.

Minor comments:

1) No error bars and confidence intervals are provided. It is standard practice to go train a network several times to determine the influence of the random seed used for training. Without such information, it is impossible to tell if the results are spurious or not. The authors must provide a measure of variance of their results

2) The chosen 6 ECG abnormalities may not be the most interesting ones to predict.

Typos/Wording:

1) Line 14 and 104: "High-performance measures were obtained [...]" => "High performance measures were obtained [...]"

2) Caption of Table 3: ")" slipped in at the end

References:

- [1] A Novel Approach for Detection of Myocardial Infarction from ECG Signals of Multiple Electrodes, IEEE Sensors Journal PP(99), DOI: 10.1109/JSEN.2019.2896308
- [2] A deep neural network learning algorithm outperforms a conventional algorithm for emergency department electrocardiogram interpretation., J Electrocardiol. 2019 Jan - Feb;52:88-95. doi: 10.1016/j.jelectrocard.2018.11.013.
- [3] Artificial intelligence to predict needs for urgent revascularization from 12-leads electrocardiography in emergency patients, PLoS One. 2019; 14(1): e0210103.
- [4] Hannun AY, et al. (2019) Cardiologist-level arrhythmia detection and classification in ambulatory electrocardiograms using a deep neural network. Nature Medicine 25(1):65–69.
- [5] Maas, A. L., Hannun, A. Y., & Ng, A. Y. (2013, June). Rectifier nonlinearities improve neural network acoustic models. In Proc. ICML (Vol. 30, No. 1, p. 3).
- [6] Klambauer, G., Unterthiner, T., Mayr, A., & Hochreiter, S. (2017). Self-normalizing neural networks. In Advances in Neural Information Processing Systems (pp. 972-981).
- [7] Goodfellow, I., Bengio, Y., Courville, A., & Bengio, Y. (2016). Deep learning (Vol. 1). Cambridge: MIT press.

Dear Editor,

We sincerely thank all reviewers for their constructive criticisms and valuable comments. The manuscript has been modified attempting to address these. Additions that we think to be relevant to the reviewers were highlighted in blue. Parts of the text that were already there in the first submission but are relevant to our response were highlighted in olive green.

We answer to reviewers comments bellow.

Best regards,

The Authors

Reviewer #1

This is a very interesting paper documenting ability of a deep neural network (DNN) to identify abnormalities in a standard 12-lead ECG recordings. The authors have used the largest to date dataset with over 2 million ECGs to train the network, subsequently validating it on about 50,000 ECGs and next testing it on a set of 827 ECGs read by health professionals with different level of expertise. The performance of the DNN was remarkable with the F1 scores (reflecting precision and recall) of 0.85 and specificity index of 0.99.

The methodological approach with so called "end-to-end" automatic classification is what mimics human approach to an ECG image, unlike few prior studies employing ECG signal processing to identify parameters of interest which subsequently were used for automatic classification.

The study employed the state-of-the-art the University fo Glasgow ECG analysis program in combination with Minnesota codes to annotate training and validation sets. The study focuses on the key clinical ECG abnormalities relevant for daily practice with practical applicability of the system for widespread use in clinics without access to ECG expertise. Although overreading of complex cases is needed, the diagnostic yield of the DNN system is very impressive and deserves implementation on daily practice.

Response to Reviewer #1

We thank the reviewer for the positive feedback. We share with him the excitement of possible daily practical applications of such technology.

Reviewer #2

General comments:

The authors introduce a novel dataset consisting of over 2 million EKG traces, using which they develop a 9-layer ResNet architecture to classify cardiac abnormalities. The performance of that model is presented and claimed to outperform medical students.

Major comments:

Results:

Ref 15 is a poor comparison for this work, apart from the fact that there are a number of questionable and arbitrary design decisions incorporated in that model, it was, similar to this paper designed and trained using a closed source dataset. The authors highlight the 2017 physionet challenge| a reputable venue

for such a class of problems, however none of the work are referenced. Notably some of the top ranked work out of that competition use Deep CNN for classifying arrhythmia, similar to this work. I urge the authors to consider the work resulting from that rigorous validation over ref 15 for the motivation for the proposed architecture.

<https://physionet.org/challenge/2017/sources/>

Hong, S., Wu, M., Zhou, Y., Wang, Q., Shang, J., Li, H. and Xie, J., 2017, September. ENCASE: An ENsemble CLASsifiEr for ECG classification using expert features and deep neural networks. In 2017 Computing in Cardiology (CinC) (pp. 1-4). IEEE.

Zabihi, M., Rad, A.B., Katsaggelos, A.K., Kiranyaz, S., Narkilahti, S. and Gabbouj, M., 2017, September. Detection of atrial fibrillation in ECG hand-held devices using a random forest classifier. In 2017 Computing in Cardiology (CinC) (pp. 1-4). IEEE.

Kamaleswaran, R., Mahajan, R. and Akbilgic, O., 2018. A robust deep convolutional neural network for the classification of abnormal cardiac rhythm using single lead electrocardiograms of variable length. *Physiological measurement*, 39(3), p.035006.

Methods:

- An extensive methodological approach is provided for both the training and testing datasets. The training dataset uses a method for automatic classification developed by one of the co-authors (PWM), i.e. the Uni-G. Why was an external EKG classification tool not compared? Can you provide some justification for the significant reliance on this software for label generation?
- There should be better empirical reasoning for why 98% of the dataset was used for training, furthermore why 827 samples were selected for the test.
- There needs to be more details highlighted about the distribution of these classes in the training, validation and test sets.
- While lines 271 through 282 reflect highly optimistic views of the work, there should be some caution when issuing states such as 'state of the art, leading to widespread use etc.' The authors are certainly commended for the work in developing the dataset, and for demonstrating this 'proof of concept' study, however due to a lack of external validation with other arrhythmia detection models these broad statements are premature.

Minor comments:

- Line 300, does a 'team of cardiologist' actually interpret the EKG or is it a single cardiologist? How many inputs are there in the final assessment? This is unclear from the current text.
- Line 334, additional clarity is required, how was the test set labeled? Were there any further validation performed on the classification model? Inclusion of this detail would be great for supplemental material.
- Line 436 Why was zero padding used, were other padding methods experimented?
- Line 447, again supplemental materials on the selection of these hyperparameters would be expected.

Response to Reviewer #2

We thank the reviewer for raising these relevant points. We answer to them bellow:

#1 - 'The authors introduce a novel dataset consisting of over 2 million EKG traces, using which they develop a 9-layer ResNet architecture to classify cardiac abnormalities. The performance of that model is presented and claimed to outperform medical students.'

We would like to highlight the fact that our algorithm outperformed medical students **and** medical residents. Medical residents, while still in training, have already finished medical school and are already certified as doctors. Summing up the 6 years of medical school and their time in residence, medical residents have already undertaken 9 to 10 years of medical training.

#2 - ‘‘Ref 15 is a poor comparison for this work, apart from the fact that there are a number of questionable and arbitrary design decisions incorporated in that model, it was, similar to this paper designed and trained using a closed source dataset. The authors highlight the 2017 physionet challenge| a reputable venue for such a class of problems, however none of the work are referenced. Notably some of the top ranked work out of that competition use Deep CNN for classifying arrhythmia, similar to this work. I urge the authors to consider the work resulting from that rigorous validation over ref 15 for the motivation for the proposed architecture.’’

We agree with the reviewer that the relevance of the 2017 physionet challenge might be underestimated in our initial submission. We have now included a phrase mentioning the performance of convolutional networks in the challenge, namely: ‘‘Furthermore, in the 2017 Physionet challenge (16), algorithms for detecting AF have been compared in an open dataset of single lead ECGs and, both the architecture described in (15) and other convolutional architectures (24, 25) have achieved top scores.’’. However, we would also like to emphasize that Ref 15 does report the performance of its architecture in the Physionet challenge and does achieve top scores. In our previous submission we already included a note about that - see lines 26 to 29.

Ref 15 is a high impact paper published in a respected journal. Hence, we believe it is important that our paper highlight similarities and differences with that work. Specifically because of the incremental nature of our paper, where we extend the use of DNNs to new scenarios and bring the technology closer to clinical practice.

Finally, we do believe that there is some degree of arbitrariness to ours (and theirs) decisions. Nevertheless, both neural architectures are the result of good deep learning practices developed empirically over the last decade [GBC16]. The residual network has been used extensively in image classification and many choices in our (and theirs) architecture are inspired by this line of development.

#3 - ‘‘An extensive methodological approach is provided for both the training and testing datasets. The training dataset uses a method for automatic classification developed by one of the co-authors (PWM), i.e. the Uni-G. Why was an external EKG classification tool not compared? Can you provide some justification for the significant reliance on this software for label generation?’’

Other softwares for automatic ECG interpretation are proprietary software and thus not available to us free of charge. We choose to use the University of Glasgow software because it is a well established automatic analysis program that has been tested and improved over the three last decades. Furthermore, we have access to it via our scientific partnership.

We would also like to highlight the fact that this software is not directly used for generating the test dataset and the final classification is based entirely on the decisions of certified cardiologists. For the training and validation set, it is used for cleaning and improving the quality of our labelling, avoiding possible mistakes by the cardiologists.

#4 - ‘‘There should be better empirical reasoning for why 98% of the dataset was used for training, furthermore why 827 samples were selected for the test.’’

This is indeed a valid and natural question. The existence of very large datasets has changed the way datasets are usually split into training, validation and test set. Traditional ways of splitting the dataset such as 50%-25%-25% (as proposed, for instance, in [FHT01]) have changed for some tasks due to the increasing availability of data. For instance, in [NK18, Ng], it is suggested that for big datasets a 98%-1%-1% split might be a reasonable choice.

This trend is reflected in many standard deep learning benchmarks. For instance, the Wikitext-103 dataset used for language modeling [MXBS16], has a split of 103,227,021 - 217,646 - 245,569 tokens which correspond to a 99.5%-0.21%-0.24% split. Our choice of the 98%-2% split between training and validation datasets follows this trend.

The test set, on the other hand is smaller because it require a more demanding and labor intensive annotation procedure. However, it is worth highlighting the fact that this test set, while small compared to our training set, has a size that is comparable to other test sets used in the literature: - Ref 15 uses a test set of 328 ECG records from different patients (40% of the size of our dataset) - Ref 33 uses 575 ECG records from different patients (70% of the size of our dataset) - The 2017 physionet challenge has a testset of 3658 ECGs from different patients (4.4 times the size of our dataset). However, it does not use a dataset annotated by multiple specialists.

#5 - ‘‘There needs to be more details highlighted about the distribution of these classes in the training, validation and test sets.’’

We thank the referee for this comment. We have this information on Table 3 in the Supplementary Material. There we included information about the distributions of the classes in training, validation and test sets. And also information about gender and age distributions.

#6 - ‘‘While lines 271 through 282 reflect highly optimistic views of the work, there should be some caution when issuing states such as ‘state of the art, leading to widespread use etc.’ The authors are certainly commended for the work in developing the dataset, and for demonstrating this ‘proof of concept’ study, however due to a lack of external validation with other arrhythmia detection models these broad statements are premature.’’

Thank you for the suggestion. We agree and, in the new version, avoid these broad and overly optimistic statements. The last paragraph of the discussion section has now been rewritten as: *“In conclusion, we developed an end-to-end DNN capable of accurately recognizing six ECG abnormalities in S12L-ECG exams, with a diagnostic performance at least as good as medical residents and students. This study shows the potential of this new technology, which, when fully developed, might lead to more reliable automatic diagnosis and improved clinical practice. Although expert review of complex and borderline cases seems to be necessary even in this future scenario, the development of such automatic interpretation by a DNN algorithm may expand the access of the population to this basic and useful diagnostic exam.”*

#7 - ‘‘Line 300, does a ‘team of cardiologist’ actually interpret the EKG or is it a single cardiologist? How many inputs are there in the final assessment? This is unclear from the current text.’’

Indeed, this was unclear from the description in the initial submission. There is a team of cardiologists on duty, but a single cardiologist analyzes each ECG. We clarify that in the new submission as follows: *“Once there, one cardiologist from the TNMG experienced team analyzes the exam and a report is made available to the health service that requested the exam through an online platform.”*

#8 - ‘‘Line 334, additional clarity is required, how was the test set labeled? Were there any further validation performed on the classification model? Inclusion of this detail would be great for supplemental material.’’

Good suggestion. The test set was labeled by a certified cardiologist who was presented with the free-text and was required to choose among the pre-specified classes. We have included this information in the new submission, viz: *“The classification model reported above was tested on 4557 medical reports manually labeled by a certified cardiologist who was presented with the free-text and was required to choose among the pre-specified classes”*

Furthermore we would like to highlight that: i) for the test set, the diagnosis is codified directly into the classes and there is no need to extract the labels from the textual report, removing this source of error from test data; and, also, ii) the annotation procedure described in Methods section C (“Training and validation set annotation”) try to mitigate possible errors arising from labelling from the text report.

#9 - ‘‘Line 436 Why was zero padding used, were other padding methods experimented?’’

The length of our ECG readers range from 7 to 10 seconds and we need to make them uniform in size for efficient computation. Zero padding is very commonly used in preprocessing images for deep learning. Furthermore, using zero padding is already the standard way of matching the size of consecutive residual neural network layers. Hence, zero padding appears as a very natural design choice to match the size of different ECG tracings. We have not tried out other alternatives since we do not believe this to be a bottleneck of our design.

#10 - ‘‘Line 447, again supplemental materials on the selection of these hyperparameters would be expected.’’

Good suggestion, thank you. We have now included a subsection in hyperparameter tuning in Methods section.

Reviewer #3

Summary:

The authors present a neural network that can classify a range of ECG abnormalities from short 12-lead ECG recordings and they compare their results with trained individuals.

Major comments:

1) The authors claim in the abstract that "The analysis of the digital ECG [...] have not been studied in an end-to-end machine learning scenario", This is a vast overstatement, as a lot of prior literature exists on this subject. The authors do soften their claims in the introductory section, where they merely claim that "it is still an open question if the application of this technology would be useful in a realistic clinical setting, where 12-lead ECGs are the standard technique". It is worth pointing out that a number of other published work has applied Deep Learning to 12-lead ECG (albeit some of this might be considered concurrent work) [1-3]. However, the dataset proposed in this study is of larger size than in [1-3], which is why I think the current work is of interest once the other comments are addressed.

2) The authors raise the biggest concern of the underlzing work themselves: "While we obtained the highest F1 scores for the DNN, the McNemar statistical test and bootstrapping suggest that we do not have confidence enough to assert that the DNN is actually better than the medical residents and students with statistical significance." (line 224)

3) There is a very high discrepancy between number of training and test samples (0.03%). A typical split would be 8:1:1 for training, validation and test. Having an additional test set to compare to experts is desirable but currently this set is tiny and the values in the results tables only stem from one cardiology resident , one emergency resident and one medical student. Also note that the F1 scores in Table 1 are on average higher than these of the emergency resident. The authors should provide more details on that.

4) There is an automatic system in place which provides Minnesota codes.

The result of this system should be used as a baseline comparison method.

5) Insufficient description of the model selection process.

The authors used a very similar network architecture as in [4] but they do not describe how they selected their final network. In particular, it is unclear which hyperparameters were considered, e.g. which learning rates, optimizers, activation functions (Relu, leaky [5], SELU [6]), number of neurons, number of layers, kernel sizes, etc were considered at all. Furthermore, the authors do not describe how the hyperparameters were selected. Has this been done by manual hyperparameter selection, grid-search or by model-based approach (see hyperparameter selection strategies given in ref [7])? The authors should provide details, which hyperparameters were considered and how they were selected together with performance comparisons.

Minor comments:

1) No error bars and confidence intervals are provided. It is standard practice to go train a network several times to determine the influence of the random seed used for training. Without such information, it is impossible to tell if the results are spurious or not. The authors must provide a measure of variance of their results

2) The chosen 6 ECG abnormalities may not be the most interesting ones to predict.

Typos/Wording:

1) Line 14 and 104: "High-performance measures were obtained [...]" => "High performance measures were obtained [...]"

2) Caption of Table 3: ")" slipped in at the end

References:

[1] A Novel Approach for Detection of Myocardial Infarction from ECG Signals of Multiple Electrodes, IEEE Sensors Journal PP(99), DOI: 10.1109/JSEN.2019.2896308

[2] A deep neural network learning algorithm outperforms a conventional algorithm for emergency department electrocardiogram interpretation., J Electrocardiol. 2019 Jan - Feb;52:88-95. doi: 10.1016/j.jelectrocard.2018.11.013.

[3] Artificial intelligence to predict needs for urgent revascularization from 12-lead electrocardiography in emergency patients, PLoS One. 2019; 14(1): e0210103.

[4] Hannun AY, et al. (2019) Cardiologist-level arrhythmia detection and classification in ambulatory electrocardiograms using a deep neural network. Nature Medicine 25(1):65{69.

[5] Maas, A. L., Hannun, A. Y., & Ng, A. Y. (2013, June). Rectifier nonlinearities improve neural network acoustic models. In Proc. ICML (Vol. 30, No. 1, p. 3).

[6] Klambauer, G., Unterthiner, T., Mayr, A., & Hochreiter, S. (2017). Self-normalizing neural networks. In Advances in Neural Information Processing Systems (pp. 972-981).

[7] Goodfellow, I., Bengio, Y., Courville, A., & Bengio, Y. (2016). Deep learning (Vol. 1). Cambridge: MIT press.

Response to Reviewer #3

We thank the reviewer for raising these relevant points. We use this letter to clarify and highlight the modifications of our re-submission:

#11 - ‘‘The authors claim in the abstract that "The analysis of the digital ECG [...] have not been studied in an end-to-end machine learning scenario", This is a vast overstatement, as a lot of prior literature exists on this subject. The authors do soften their claims in the introductory section, where they merely claim that "it is still an open question if the application of this technology would be useful in a realistic clinical setting, where 12-lead ECGs are the standard technique". It is worth pointing out that a number of other published work has applied Deep Learning to 12-lead ECG (albeit some of this might be considered concurrent work) [1-3]. However, the dataset proposed in this study is of larger size than in [1-3], which is why I think the current work is of interest once the other comments are addressed.’’

Our intention was to refer to 12-lead ECGs and to be read in the context of the previous statement (‘‘We present a Deep Neural Network (DNN) model for predicting electrocardiogram (ECG) abnormalities in short-duration 12-lead ECG recordings.’’). However, we do see your point and we agree that the statement might lead to confusion. Hence, we have now removed this claim from the abstract altogether.

We included the 3 references [1-3] in our citation list (Refs 30, 31 and 38) and the differences between theirs and our work is now reviewed in the Discussion session.

#12 - ‘‘There is a very high discrepancy between number of training and test samples (0.03%). A typical split would be 8:1:1 for training, validation and test. Having an additional test set to compare to experts is desirable but currently this set is tiny and the values in the results tables only stem from one cardiology resident , one emergency resident and one medical student.’’

We have addressed a similar question from Reviewer 2 in comment #4. We kindly ask the referee to take a look at our answer there. The two main points were that: i) the existence of very large datasets has changed the way datasets are usually split into training, validation and test sets [3, 4]; and, ii) the size of the test set, while small compared to our training set, is comparable in size to other test sets used in the literature.

Besides that, we would like to emphasize that we are using for comparison **two** cardiology residents, **two** emergency resident and **two** medical students. By using two of each of each class, each one annotating half of the dataset, we intend to average out, to some extent, the individual skill of each annotator.

#13 - ‘‘... Also note that the F1 scores in Table 1 are on average higher than these of the emergency resident. The authors should provide more details on that.’’

Thank you for the suggestion. The performance of the emergency residents is indeed worse than medical students for many abnormalities and this is counter-intuitive since they have less medical training. We included a phrase interpreting this result in the discussion: *‘‘Interestingly, the performance of the emergency residents is worse than medical students for many abnormalities. This might seem counter-intuitive because they do have less years of medical training. It might, however, be justified by the fact that emergency residents, unlike cardiology residents, do not have to interpret these exams on a daily basis, while medical students still have these concepts fresh from their studies.’’*

#14 - ‘‘There is an automatic system in place which provides Minnesota codes. The result of this system should be used as a baseline comparison method.’’

The University of Glasgow software is indeed a very natural baseline for our performance. Table I in the present document compares the performance of the DNN with University of

	Precision (PPV)		Recall (Sensitivity)		Specificity		F1 Score	
	DNN	Uni-G	DNN	Uni-G	DNN	Uni-G	DNN	Uni-G
1dAVb	0.867	0.742	0.929	0.821	0.995	0.990	0.897	0.780
RBBB	0.895	0.938	1.000	0.882	0.995	0.997	0.944	0.909
LBBB	1.000	1.000	1.000	0.767	1.000	1.000	1.000	0.868
SB	0.833	0.722	0.938	0.812	0.996	0.994	0.882	0.765
AF	1.000	0.524	0.769	0.846	1.000	0.988	0.870	0.647
ST	0.947	0.875	0.973	0.946	0.997	0.994	0.960	0.909

TABLE I. Performance on the test set. Scores of our DNN are compared with University of Glasgow automatic diagnosis. (PPV = positive predictive value)

Glasgow software. While the DNN do outperform Glasgow by a safe margin, we are very reluctant to include such analysis in the text and would refrain to do so, unless the reviewers insist on that point (in that case we would be willing to include that in the final version).

Our experiment was not designed to evaluate a commercial software, and we would like to refrain from publishing about Glasgow software errors in a setup for which the software was not entirely designed for and draw unintended conclusions. Specially because other commercial software are not available to us, as mentioned in question #3, and cannot be evaluate under the same standards. While our comparison is fair, highly standardized and do reflect what type of ECG arrives daily at TNMG for analysis, it does contain borderline cases (cf. discussion on lines 223-232 and Fig. 5) and exams partially corrupted with noise (cf. Table 2). Also, practical conditions faced at TNMG - such as the existence of exams with 7 seconds from the Brazilian ECG machine of the brand Micromed - might not have been completely accounted for during the design of University of Glasgow software and may have degraded its performance. The error rates under these conditions might be higher than in setups where commercial software usually report their performance on and might draw unintended conclusions from the reader.

#15 - ‘‘Insufficient description of the model selection process. The authors used a very similar network architecture as in [4] but they do not describe how they selected their final network. In particular, it is unclear which hyperparameters were considered, e.g. which learning rates, optimizers, activation functions (Relu, leaky [5], SELU [6]), number of neurons, number of layers, kernel sizes, etc were considered at all. Furthermore, the authors do not describe how the hyperparameters were selected. Has this been done by manual hyperparameter selection, grid-search or by model-based approach (see hyperparameter selection strategies given in ref [7])? The authors should provide details, which hyperparameters were considered and how they were selected together with performance comparisons.’’

Thank you for the suggestion. We have now included a subsection about hyperparameter tuning in Methods sections in order to address your question and Comment #10 from the Reviewer #2.

We refrain from including a direct performance comparison since the choice of architecture and hyperparameters was done in parallel with improvements in the dataset. Hence, to directly compare the performance of all tested hyperparameters would require us to reevaluate all hyperparameters on the newest version of the dataset. This would be computationally very expensive and, furthermore, it would not be consistent with the methodology we have actually use to choose the hyperparameters.

We explicitly mention in the text that *‘‘the choice of architecture and hyperparameters was done together with improvements in the dataset’’*.

#16 - ‘‘No error bars and confidence intervals are provided. It is standard practice to go train a network several times to determine the influence of the random seed used for training. Without such information, it is impossible to tell if the results are spurious or not. The authors must provide a measure of variance of their results.’’

Thank you for the suggestion. We have trained the neural network 10 times for our new submission. Figure 1 now include both the result of the nominal model (strong line) and

a shaded region including the worst and best precision (for each threshold) among the 10 realizations. The nominal model, now used in all the rest of the paper, is the one with a intermediate performance among the 10 realizations (more precisely, the one with micro average precision immediatly above the median of all realizations).

The folowing paragraph was added in the text to describe this analysis: “*Neural networks are initialized randomly, and different initialization usually yield different results. In order to show the stability of the method, we have trained 10 neural networks with the same set of hyperparameters and different initializations. The range between the maximum and minimum precision among these realizations, for different values of thereshold, are the shaded regions displayed in Figure 1. These realizations have micro average precision (mAP) between 0.946 and 0.961, we choose the one with mAP immediatly above the median value of all executions (the one with mAP = 0.951). All the analysis from now on will be for this realization of the neural network, which correspond both to the strong line in Figure 1 and to the scores presented in Table 1. For this model, Figure 1 shows the point corresponding to the maximum F1 score for each abnormality. The threshold corresponding to this point is used for producing the DNN scores displayed in Table 1.*”

And the following footnote describe why we picked the value immediatly above the median: “*We couldn’t choose the model with mAP equal to the median value because 10 is even number, hence there is no single middle value.*”

#17 - ‘‘The chosen 6 ECG abnormalities may not be the most interesting ones to predict.’’

We choose those abnormalities that we considered to be representative of both rhythmic and morphological abnormalities. We believe this to be an interesting choice for our proof-of-concept study, even though alternative choices might also have been possible. The inclusion of more classes was cited as a future direction of our work.

References

- [FHT01] Jerome Friedman, Trevor Hastie, and Robert Tibshirani. *The Elements of Statistical Learning*, volume 1. Springer series in statistics New York, 2001.
- [GBC16] Ian Goodfellow, Yoshua Bengio, and Aaron Courville. *Deep Learning*. MIT Press, 2016.
- [MXBS16] Stephen Merity, Caiming Xiong, James Bradbury, and Richard Socher. Pointer Sentinel Mixture Models. *arXiv:1609.07843*, September 2016.
- [Ng] Andrew Y. Ng. Practical aspects of Deep Learning (Coursera). <https://www.coursera.org/lecture/deep-neural-network/train-dev-test-sets-cxG1s>.
- [NK18] Andrew Y. Ng and Kian Katanforoosh. CS230 Deep Learning (Stanford). <https://cs230-stanford.github.io/train-dev-test-split.html>, January 2018.

Reviewers' comments:

Reviewer #2 (Remarks to the Author):

The authors have satisfactorily addressed most of the revisions, however my concern with the low numbers in the test-set is still a factor. A follow-up validation set may be warranted to truly establish the quality of the proposed model and pipeline.

Reviewer #3 (Remarks to the Author):

Ad comment 11

The authors claim that they have deleted following sentence:

"The analysis of the digital ECG obtained in a clinical setting can provide a full evaluation of the cardiac electrical activity and have not been studied in an end-to-end machine learning scenario."

This sentence is still included in the final version (July 19, 2019) albeit marked as deleted in the annotated version. This has to be fixed.

Ad comment 12 (as well as comment 4)

The problem for me is not so much the validation set but the test set.

The authors compare with Ref 15 (Hannun AY, et al. (2019)) but there they have at least a distinction between sequence-level ($n=7,544$) and set-level ($n=328$), where set-level describes the unique set of algorithm predictions that are present in the 30-s record. This alleviates the problem at least

a little bit and I also think - like Reviewer #1 - that some design choices in this paper were questionable.

The authors have in their test set for example of abnormality AF only 13 positive cases and for SB only 16 positive cases (see Table 5).

With that underlying data it is of course hard to get any significant results.

I am very well aware of the fact that two people of each group were involved, still it would have been interesting to see if there would have been some disagreements within the group to be able to get a better feeling of the differences of skill sets.

Anyway, properly annotating data costs money and the authors have at least corrected their overly optimistic claims (e.g. response to comment 6).

Ad comment 14

With the baseline results from Uni-G I did get a much better feeling for the data, which would be a good selling point for the paper. I would at least include these results in the Supplement.

Ad comment 15

The adaptations done are sufficient.

Be aware that there are typos in line 509 (tunning  tuning; white space in front of comma).

Ad comment 16

This looks much better now.

We sincerely thank the reviewers and editor for their continuing effort to improve the paper. Additions and modifications that we think might be relevant to the reviewers are highlighted in blue in the new version of the paper. We answer to reviewer comments below.

Best regards,

The authors

Editor

Please note that in addition to our reviewers' feedback pasted below, we have sought further advice from reviewer #3 on how the concerns raised by both reviewers should be addressed. Following reviewer #3's advice, we would ask that you please provide additional experiments where your model is trained on 90% of the data, validated on 5% of the data, and performance is reported on the remaining 5% test data. A stratified split in which all traces from one individual go together in one of the folds or splits should be used. Additionally, a time-split in which the traces are sorted by their date of recording (with the 90% earlier traces being used for training, the next 5% for validation, and the newest 5% for testing) should be used so that a prospective use of the method can be simulated. Finally, inter-rater reliability should be reported for the comparison of human and model performance on the final set of 827 traces.

Response to Editor

```
§1 - ‘‘Please note that in addition to our reviewers’ feedback pasted below, we have sought further advice from reviewer #3 on how the concerns raised by both reviewers should be addressed. Following reviewer #3’s advice, we would ask that you please provide additional experiments where your model is trained on 90% of the data, validated on 5% of the data, and performance is reported on the remaining 5% test data. A stratified split in which all traces from one individual go together in one of the folds or splits should be used. Additionally, a time-split in which the traces are sorted by their date of recording (with the 90% earlier traces being used for training, the next 5% for validation, and the newest 5% for testing) should be used so that a prospective use of the method can be simulated.’’
```

While we find the analysis interesting, we also would like to highlight that the procedure of using multiple test set is not standard. The analysis of the results is indeed not trivial, since the gold standard is not sufficiently reliable on the 5% split used for testing. Hence, there is a performance drop when comparing the 5% split used as secondary test set and the original, and more reliable, test set ($n=827$). Manual revision from a certified cardiologist, however, indicated that the neural network is correct (and not the actual gold standard) in a high percentage ($\approx 80\%$) of the cases on the 5% test set for which the neural network have been considered wrong. In the end, testing the accuracy in a dataset in which the gold standard has not been reviewed has limited value, but we hope it helps addressing the reviewers and editor concerns.

We added one paragraph to the body of the text (on the Results section): *Finally, to assess the effect of how we structure our problem, we have considered alternative scenarios where we use the 2,322,513 ECG records in 90%-5%-5% splits, stratified randomly, by patient or in chronological order. Being the splits used, respectively, for training, validation and as a second larger test set. The results indicate no statistically significant difference between the original DNN used in our analysis and the alternative models developed in the 90%-5%-5% splits. The exception is the model developed using the chronologically order split, for which the changes along time in the telehealth center operation have affected the splits (cf. Figure 6).*

And one to the Methods section: *We evaluate the F_1 score for alternative scenarios where we use 90%-5%-5% splits of the 2,322,513 records. With the splits ordered: randomly; by date;*

and, stratified by patients. The neural networks developed in these alternative scenarios are evaluated on both the original test set ($n=827$) and on the additional test splits (last 5% split). The distribution of the performance in each scenario is computed by a bootstrap analysis (with 200 samples) and the resulting boxplots are displayed in the supplementary material.

The remaining analysis is relegated to the caption of Figure 6. Since the analysis is non-standard, not entirely trivial and lengthy, we think this choice helps to keep the text clear and easy to understand.

Figure 1. Boxplots for the bootstrapped F_1 scores for the DNN using alternative 90%-5%-5% splits for training, validation and as a *secondary* test set. For the splits ordered: randomly; by date; and, stratified by patients. In all cases, the performance is evaluated on: (a) the original test set ($n = 827$); and, on (b) the secondary test set (last 5% split). On (a), we also present the original DNN performance for comparison, which was developed using a 98%-2% split. The performance gap between (a) and (b) is due to the difference in the gold standard. The secondary test set obtained from the last 5% has a less accurate annotation, since it has not been annotated by multiple doctors and it uses natural language processing to extract the diagnosis from a written report. This extra noise result in worse F_1 score in (b) when compared with (a). On the other hand, the secondary 5% test split contain more than 100,000 records, which yield more stable performance in the bootstrap analysis, with more concentrated empirical distributions for the F_1 score. Both RBBB and LBBB (highlighted on the plot) present on (b) a statistically significant difference between the performance of the split ordered by date and the other two splits, that difference is due to some changes in personal that took place in the the Telehealth center, that affected the period used in the test set (10-2016 to 06-2017), resulting in lower annotation quality. A certified cardiologist reviewed cases for which the neural network have been considered wrong when compared to the gold standard from the 5% split collected from 10-2016 to 06-2017, 100 supposedly wrong RBBB and 100 supposedly wrong LBBB. The certified cardiologist reported that the neural network is actually correct, respectively, 86% and 83% percent of the cases. This analysis show the importance of a test set with a good annotation quality to obtain reliable estimation of the DNN performance. And, also, that periods of lower annotation quality in the dataset are overcome by a very high number of examples, allowing our method to yield a highly accurate model, despite the presence of such periods.

§2 - ‘‘Finally, inter-rater reliability should be reported for the comparison of human and model performance on the final set of 827 traces’’

We added Table 6(b), containing the inter-rater reliability comparing human and model on the final set of 827 traces.

Reviewer #2

The authors have satisfactorily addressed most of the revisions, however my concern with the low numbers in the test-set is still a factor. A follow-up validation set may be warranted to truly establish the quality of the proposed model and pipeline.

Response to Reviewer #2

§3 - ‘‘The authors have satisfactorily addressed most of the revisions, however my concern with the low numbers in the test-set is still a factor. A follow-up

validation set may be warranted to truly establish the quality of the proposed model and pipeline.’’

Following a direct recommendation from the Editor in this regard (cf. §1), we have included the analysis on alternative 95%-5%-5% splits (ordered randomly, by date and stratified by patient). Where 90% is used for training, 5% is used for validation and 5% as a secondary test set. Please take a look at figure 6(b). We think the requested analysis has some limitations, but we hope it helps addressing this concern.

Reviewer #3

Ad comment 11

The authors claim that they have deleted following sentence:

"The analysis of the digital ECG obtained in a clinical setting can provide a full evaluation of the cardiac electrical activity and have not been studied in an end-to-end machine learning scenario."

This sentence is still included in the final version (July 19, 2019) albeit marked as deleted in the annotated version. This has to be fixed.

Ad comment 12 (as well as comment 4)

The problem for me is not so much the validation set but the test set. The authors compare with Ref 15 (Hannun AY, et al. (2019)) but there they have at least a distinction between sequence-level (n=7,544) and set-level (n=328), where set-level describes the unique set of algorithm predictions that are present in the 30-s record. This alleviates the problem at least a little bit and I also think - like Reviewer #1 - that some design choices in this paper were questionable.

The authors have in their test set for example of abnormality AF only 13 positive cases and for SB only 16 positive cases (see Table 5). With that underlying data it is of course hard to get any significant results.

I am very well aware of the fact that two people of each group were involved, still it would have been interesting to see if there would have been some disagreements within the group to be able to get a better feeling of the differences of skill sets.

Anyway, properly annotating data costs money and the authors have at least corrected their overly optimistic claims (e.g. response to comment 6).

Ad comment 14

With the baseline results from Uni-G I did get a much better feeling for the data, which would be a good selling point for the paper. I would at least include these results in the Supplement.

Ad comment 15

The adaptations done are sufficient.

Be aware that there are typos in line 509 (tunning  tuning; white space in front of comma).

Ad comment 16

This looks much better now.

Response to Reviewer #3

§4 - ‘‘The authors claim that they have deleted following sentence: ‘‘The analysis of the digital ECG obtained in a clinical setting can provide a full evaluation of the cardiac electrical activity and have not been studied in an end-to-end machine learning scenario.’’ This sentence is still included in the final version (July 19, 2019) albeit marked as deleted in the annotated version. This has to be fixed.’’

Thank you for pointing this out. We have just fixed it.

§5 - ‘‘The problem for me is not so much the validation set but the test set. The authors compare with Ref 15 (Hannun AY, et al. (2019)) but there they have at least a distinction between sequence-level (n=7,544) and set-level (n=328), where set-level describes the unique set of algorithm predictions that are present in the 30-s record. This alleviates the problem at least a little bit and I also think - like Reviewer #1 - that some design choices in this paper were questionable.

The authors have in their test set for example of abnormality AF only 13 positive cases and for SB only 16 positive cases (see Table 5). With that underlying data it is of course hard to get any significant results.

I am very well aware of the fact that two people of each group were involved, still it would have been interesting to see if there would have been some disagreements within the group to be able to get a better feeling of the differences of skill sets.

Anyway, properly annotating data costs money and the authors have at least corrected their overly optimistic claims (e.g. response to comment 6).’’

We answer here as in §3, where the Reviewer #2 also expressed concern with the size of the test set. Following a direct recommendation from the Editor (cf. §1) in this regard, we have included the analysis on alternative 95%-5%-5% splits (ordered randomly, by date and stratified by patient). Where 90% is used for training, 5% is used for validation and 5% as a secondary test set. Please take a look at figure 6(b). We think the requested analysis has some limitations, but we hope it helps addressing this concern.

§6 - ‘‘With the baseline results from Uni-G I did get a much better feeling for the data, which would be a good selling point for the paper. I would at least include these results in the Supplement.’’

We are happy that the additional results helped with your concerns. We are willing to include the table in the final version if this is directly requested.

§7 - ‘‘The adaptations done are sufficient. Be aware that there are typos in line 509 (tunning  tuning; white space in front of comma).’’

Ok, we fixed it.

Reviewer #3 (Remarks to the Author):

My concerns have been addressed accordingly and I am confident that the paper improved compared to the initial version.

REVIEWERS' COMMENTS:

Reviewer #3 (Remarks to the Author):

My concerns have been addressed accordingly and I am confident that the paper improved compared to the initial version.

> We thank the reviewer for taking the time and effort necessary to review the manuscript.